# Feasibility of Mouth-to-Mouth Ventilation through FPP2 Respirator in BLS Training during COVID-19 Pandemic (MOVERESP Study): Simulation-Based Study

**DOI:** 10.3390/children9111751

**Published:** 2022-11-15

**Authors:** Martina Kosinová, Petr Štourač, Tereza Prokopová, Tereza Vafková, Václav Vafek, Daniel Barvík, Tamara Skříšovská, Jan Dvořáček, Jana Djakow, Jozef Klučka, Jiří Jarkovský, Pavel Plevka

**Affiliations:** 1Department of Simulation Medicine, Faculty of Medicine, Masaryk University, Kamenice 5, 625 00 Brno, Czech Republic; 2Department of Paediatric Anaesthesiology and Intensive Care Medicine, University Hospital Brno and Faculty of Medicine, Masaryk University, Jihlavská 20, 625 00 Brno, Czech Republic; 3Department of Anaesthesiology and Intensive Care Medicine, University Hospital Brno and Faculty of Medicine, Masaryk University, Jihlavská 20, 625 00 Brno, Czech Republic; 4Department of Comprehensive Cancer Care, Masaryk Memorial Cancer Institute, Faculty of Medicine, Masaryk University, Žlutý kopec 534/7, 656 53 Brno, Czech Republic; 5Paediatric Intensive Care Unit, NH Hospital Inc., 268 01 Hořovice, Czech Republic; 6Institute of Biostatistics and Analyses, Faculty of Medicine, Masaryk University, Kamenice 126/3, 625 00 Brno, Czech Republic; 7Central European Institute of Technology, Masaryk University, Kamenice 753/5, 625 00 Brno, Czech Republic

**Keywords:** resuscitation, mouth-to-mouth breathing, life support, COVID-19, training

## Abstract

Background: Due to the COVID-19 pandemic, Basic Life Support (BLS) training has been limited to compression-only or bag–mask ventilation. The most breathable nanofiber respirators carry the technical possibility for inflation of the mannequin. The aim of this study was to assess the efficacy of mouth-to-mouth breathing through a FFP2 respirator during BLS. Methods: In the cross-over simulation-based study, the medical students performed BLS using a breathable nanofiber respirator for 2 min on three mannequins. The quantitative and qualitative efficacy of mouth-to-mouth ventilation through the respirator in BLS training was analyzed. The primary aim was the effectivity of mouth-to-mouth ventilation through a breathable respirator. The secondary aims were mean pause, longest pause, success in achieving the optimal breath volume, technique of ventilation, and incidence of adverse events. Results: In 104 students, effective breath was reached in 951 of 981 (96.9%) attempts in Adult BLS mannequin (Prestan), 822 of 906 (90.7%) in Resusci Anne, and 1777 of 1857 (95.7%) in Resusci Baby. In Resusci Anne and Resusci Baby, 28.9%/15.9% of visible chest rises were evaluated as low-, 33.0%/44.0% as optimal-, and 28.8%/35.8% as high-volume breaths. Conclusions: Mouth-to-mouth ventilation through a breathable respirator had an effectivity greater than 90%.

## 1. Introduction

Cardiopulmonary resuscitation (CPR) and its features represent an essential part of education for all medical providers and first responders and should be trained repeatedly [1,2,3,4]. Following the declaration of the World Health Organization in March 2020, COVID-19 was characterized as a worldwide pandemic [5]. The European Resuscitation Council (ERC) COVID-19 guidelines emphasized the rescuer’s safety [6]. They suggested lay rescuers consider compression-only CPR and, if they are willing, trained, and able to do so, may wish to deliver rescue breaths to children in addition to chest compressions during CPR, as ventilation during CPR is strongly associated with better outcomes [7,8]. The COVID-19 pandemic may be a temporary situation, but the suggested modifications are applicable also for situations when another transmittable disease is suspected. Mouth-to-mouth ventilation remains as an integral part of Basic Life Support (BLS) in ERC BLS guidelines 2021 for both adults [9] and children [10]. However, due to the COVID-19 pandemic, BLS often had to be limited to chest compressions because of the fear of possible simulator contamination [11]. Therefore, medical students who began their studies during the pandemic, as well as other BLS-trained rescuers, lost the possibility of practical BLS training including mouth-to-mouth ventilation. Another practical option for rescue breath delivery is bag–mask ventilation. In BLS settings, this option is limited by the availability of medical equipment (self-expanding bag, face mask). Beside some specialized equipment, including that invented by Lederer and Isser for a barrier resuscitation provided by laypersons [12], respirators with optimal protection (greater than 95% FFP2 or FFP3) could yield the possibility of rescuer protection in combination with effective breath delivery.

Simulations in health care education, especially during CPR training, gained increasing importance during recent decades because of learning facilitation, CPR process quality improvement, and overall patient outcome improvement [13]. The COVID-19 pandemic has created an urgency for training while making in situ simulation challenging [14,15].

Currently, polymerase chain reaction testing, together with the ongoing vaccination process, could significantly reduce the risk of virus transmission during simulation training. Another risk is linked with the possibility of infection during the time window from the negative test to the time of simulation training. The virus particles could be detected in the nasal and throat swabs of asymptomatic humans [16]. Therefore, we aimed to discover any possible method minimizing extra manipulation with personal protective equipment to enable a safe course of practical lectures, including the BLS training in our Simulation Centre. It appears reasonable that the most efficient technique to minimize the risk of virus transmission during BLS training could be the combination of vaccination and testing together with protective mouth-to-mouth breathing through the respirator. Another advantage is that a suitable respirator can be placed on the rescuer’s face during the whole practice session with no extra manipulation needed (see Appendix A with a practical demonstration). Despite the high effectivity of filtering face piece (FFP) 2 or even FFP3 respirators, they are unable to completely prevent the virus transmission, although they are properly sealed [17]. Most conventional respirators use up to five layers of melt-blown filter material and up to five layers of FFP2 filter. Polylactic acid, polypropylene, and copper currently seem to be the most promising materials with the highest antimicrobial activity that could reduce viral viability [18,19]. Theoretically, the three-layer FFP2 nanofiber self-sterilizing respirator with accelerated copper (VK RespiRaptor^®^ FFP2 Nanofiber Respirator, Respilon, Czech Republic) could be used for mouth-to-mouth ventilation due to its unique composition and properties [20]. The three-layer mask is highly effective in inactivating SARS-CoV-2 by preventing the virus from passing through [21,22].

The primary aim of this study was to assess the effectivity of mouth-to-mouth ventilation through breathable self-sterilizing nanofiber respirators with accelerated copper, assessed by visible chest rises in three different mannequins. Secondary outcomes included the determination of mean no-flow time, longest no-flow time, success in achieving the optimal breath volume, the technique of mouth-to-mouth ventilation, and incidence of adverse events.

## 2. Materials and Methods

The cross-over simulation-based study (Mouth-to-mouth Ventilation Efficiency through breathable self-sterilizing Respirator during BLS in COVID-19 pandemic: MOVERESP study) in the Simulation Centre of the Faculty of Medicine of Masaryk University was conducted from 3 May 2021 to 15 May 2021. Participants were voluntary medical students from a training program for student lectors of first aid who fulfilled the inclusion criteria: informed consent to participate and trained in providing BLS according to ERC BLS guidelines 2021. Students from group 1 (Student as Teacher 1.0; SaT 1.0) were experienced in BLS teaching, including mouth-to-mouth ventilation. Students from group 2 (Student as Teacher 2.0; SaT 2.0) have not yet completed the teaching program but underwent prior training in adult and infant BLS, including mouth-to-mouth ventilation, during their studies before the COVID-19 pandemic. For in-person participation in education at the university campus and participation in the MOVERESP study, students need to be vaccinated or record a negative COVID-19 test checked at regular intervals according to the Ministry of Health of the Czech Republic recommendations.

Overall, 187 students were actively addressed, with 100 from the SaT 1.0 program and 87 students from the SaT 2.0 program. A total of 104 students provided informed consent and participated in the measurements. Each student provided a 2 min cycle of single-rescuer BLS according to ERC guidelines 2021, wearing the breathable nanofiber self-sterilizing respirator in three different mannequins: Professional Adult Medium Skin CPR-AED Training Manikin with CPR Monitor (Prestan), further mentioned as “Adult BLS” mannequin; Resusci Anne QCPR AED (Laerdal), further mentioned as “Resusci Anne”; and Resusci Baby QCPR (Laerdal), further mentioned as “Resusci Baby”. The mannequin sequence was randomized in a 1:1:1 allocation and students were allocated into three groups with random mannequin sequence (Figure 1). Each 2 min cycle of BLS was separated by 3 min of rest period. Respirator properties: The first and third layers are made of polypropylene non-woven fabric, which is very light and breathable and at the same time is enriched with an action method that serves as a virus inactivator. The second layer is formed by a filter containing polyacrylonitrile nanofiber. This filter has been specifically designed to achieve very high filtration efficiency (98.59% to 99.78 with sodium chloride testing and 99.06% to 99.66% with paraffin oil testing) and is also supported by low weight, 4.4 g (conventional respirators weight around 8 g and more). The achieved breathability is between 30 to 35 Pa at an airflow of 30 L min^−1^ (the standard sets 70 Pa as the maximum value for a respirator in rest mode) and 100 to 117 Pa at a flow rate of air of 95 L min^−1^ (standardized maximum 240 Pa for the hard work). An accelerated copper alloy surface may be capable of suppressing virus transmission without releasing any copper particles. The respirator had the following certifications: EU Type-Examination Certificate No.: 0200-PPE-08903 version 3 (COVID-19); EC Declaration of Conformity; Viral Filtration Test Report-Nelson Labs (VFE); EN ISO 10,993 skin irritability and hypersensitivity—The National Institute of Public Health; C&D PIP Code: 4166021. The primary outcome was defined as the global effectivity of mouth-to-mouth (mouth-to-mouth-and-nose in an infant) ventilation by No breath or Visible breath. Data from all three mannequins were analyzed for the effectivity of mouth-to-mouth ventilation. The secondary aim of the study was the quantitative breath analysis (QCPR Skill Reporter^®^) measured on Resusci Anne and Resusci Baby. The optimal breath volume was set in Resuci Anne at 400 to 600 mL and in Resusci Baby at 30 to 50 mL. For quantitative analysis of visible breaths, three breath levels were defined: low-volume breath (below the set margin of optimal breath), optimal-volume breath, and high-volume breath (over the margin of optimal breath). The follow-up questionnaire comprised two questions and space for any notes about the use of a respirator and was sent to all participants. Trained observers assessed the measurements on mannequins with QCPR Skill Reporter: the mean volume of rescue breath during a 2 min cycle of BLS and the incidence of achieving the optimal breath volume. During every single rescue breath, the presence of head tilt (in the two adult mannequins) or neutral head position (in the infant mannequin) and nose pinching (in adult mannequins) were recorded. The non-valid attempts (without the correct mouth-to-mouth/mouth-to-mouth-and-nose ventilation technique according to ERC guidelines) were excluded from the final analysis. The single observer evaluation precluded interindividual variability. The intraindividual variability was reduced by pre-study observer training. According to the Ethics Board of the Faculty of Medicine, Masaryk University, Brno (Chairperson: Michal Koščík), this type of study does not involve research subjects as it is performed exclusively on mannequins. The study is therefore exempted from ethical review under the laws of the Czech Republic or the internal regulations of the Faculty of Medicine, Masaryk University. The trial was registered on clinicaltrials.gov (NCT04867265). Written informed consent was obtained from all participating medical students. A power analysis using PASS 16 Power Analysis and Sample Size Software (2018, NCSS, LLC., Kaysville, UT, USA, https://www.ncss.com/software/pass/, accessed on 13 April 2021) was calculated (estimation of the width of the 95% confidence interval) with the sample size of *n* = 80 or higher. Randomization was based on the envelope method. Each envelope contains the order of use of three mannequins with equal probability of each possible combination. The analysis was computed both on the level of subjects and their attempts. Due to this hierarchical structure of the data, the categorical endpoints were described as absolute and relative frequencies of categories of attempts and mean relative frequencies of aggregated subjects’ data. In addition, the statistical significance of differences between mannequins was computed using a generalized linear mixed effects model. Analysis was computed using SPSS 26.0.0.0. and *p* = 0.05 was set as the level of statistical significance in all analyses. The power analysis was computed for the mixed models’ tests for two proportions with power 90%, alpha 0.05, number of measurements per cluster of 9, and the difference between proportions 90% and 95% with the result of 91 clusters in each group. Taking into account the 10% reserve, the total sample size was estimated to be 100 in each group.

## 3. Results

Overall, 104 medical students agreed to participate: 34 students (34.0%) from program SaT 1.0 and 70 students (80.5%) from SaT 2.0 (Figure 1).

The total number of performed attempts of rescue breaths (valid; non-valid; % of non-valid) was *n* = 1007 (981; 26; 2.6%) in Adult BLS mannequin, *n* = 969 (906; 63; 6.5%) in Resusci Anne, and *n* = 1908 (1857; 51; 2.7%) in Resusci Baby. Visible chest rise was detected in 96.9% (*n* = 951) of the valid attempts in Adult BLS mannequin, 90.7% (*n* = 822) in Resusci Anne, and 95.7% (*n* = 1777) in Resusci Baby (Table 1).

Comparing SaT 1.0 and SaT 2.0 in the global effectiveness of mouth-to-mouth ventilation, there was no statistically significant difference in the Adult BLS mannequin (SaT 1.0 *n* = 328, 96.65%; SaT 2.0 *n* = 653, 97.09%; *p* = 0.837) and Resusci Baby (SaT 1.0 *n* = 634, 96.37%; SaT 2.0 *n* = 1223, 95.42%; *p* = 0.120). However, a significant difference in the visibility of chest rise was detected in Resusci Anne (SaT 1.0 *n* = 282, 97.52%; SaT 2.0 *n* = 624, 87.66%; *p* = 0.008). In the secondary analysis, optimal breath volume was reached in 33% (*n* = 299) of attempts in Resusci Anne and 44% (*n* = 817) in Resusci Baby (Table 2).

The mean measured duration of pauses for ventilation between compressions (no-flow time interval) in a 2 min cycle in Resusci Anne was 7.3 s in SaT 1.0 (min. 4.0–max. 9.8), 7.4 s in SaT 2.0 (min. 5.0–max. 12.0), and 7.4 s in both groups of students combined (*n* = 104), while for Resusci Baby, the mean was 6.0 s in SaT 1.0 (min. 4.0–max. 9.8), 5.9 s in SaT 2.0 (min. 4.1–max. 13.0), and 5.9 s in both groups combined. In total, students felt the need to adjust the respirator on the face 19 times. The adjustment of the respirator was never followed by an ineffective breath. The follow-up questionnaire was filled in by 100% of respondents (Table 3, *n* = 104).

## 4. Discussion

To our knowledge, this is the first study evaluating the feasibility of mouth-to-mouth ventilation through the FFP2 respirator during BLS training in the COVID-19 pandemic. The most important finding of this randomized simulation-based study was that mouth-to-mouth ventilation through a breathable respirator with accelerated copper, assessed by the visible chest rise, was more than 90% effective in three different mannequins (Table 1).

Based on the recommendations for ventilation during rescue breathing [10], all attempts with incorrect head position or without pinched nose were excluded from the analysis. The well-established recommendation for a rescue breath [8] is to give each rescue breath more than 1 s to achieve a visible chest rise [23]. Based on this recommendation, the effectivity of a rescue breath was defined as a visible chest rise in the mannequin [10,23]. In the Adelborg study, lifeguard professionals achieved 91% effectiveness with mouth-to-mouth ventilation. In the same study, lower effectivity in mouth-to-pocket-mask ventilation (79%) and bag–mask–valve ventilation (59%) was achieved. The effectivity of mouth-to-mouth ventilation through a respirator in the MOVERESP study was comparable with previous data before the COVID-19 pandemic without using a respirator [24]. The highest effectivity (96.9%) of rescue breaths was achieved in the Adult BLS mannequin, widely used in both the Czech Republic and internationally for BLS training. This shows the high potential for worldwide use on one of the standard BLS mannequins. However, further testing on other BLS mannequin types might be necessary since there were differences between the used mannequins (Table 1).

The only statistically significant difference in the effectiveness of mouth-to-mouth ventilation depending on the practitioner’s experience was in Resusci Anne (SaT 1.0 97.52% vs. SaT 2.0 87.66%; *p* = 0.008). Moreover, there were more extreme values in the mean duration of pauses in the SaT 2.0 program. We assume that the more profound training and actual experience in teaching BLS in SaT 1.0 makes a difference between students from the two programs.

The duration of no-flow time has a significant impact on the patient’s outcome. Therefore, the duration of pauses between the compressions was defined as one of the secondary aims, as this could be prolonged due to difficulties with the use of a respirator for mouth-to-mouth ventilation. Pause time for mouth-to-mouth ventilation in the Adelborg study was 8.9 s on average. In a study regarding ventilation during CPR, the median no-flow time of the dispatched first responders was 7 s (25th–75th percentile, 5–9 s) and 8 s (25th–75th percentile, 7–10 s; *p* = 0.059) for the onsite lay rescuers [25]. These results are comparable with a mean pause time of 7.4 s in Resusci Anne and 5.9 s in Resusci Baby in the MOVERESP study, which is why the breathable respirator should not prolong the pauses between compressions.

Standardly available five-layer FFP2 or FFP3 respirators are, based on our experience, not technically suitable for administering effective breath due to the twice lower breathability of the used three-layer respirator. When comparing the mean volume of each student, the majority of values were within the optimal breath volume range or higher in both Resusci Anne and Resusci Baby (Figure 2), which is also comparable with the results from the Adelborg study [24]. In contrast, when asking the students, they felt the need for a bigger breath as they thought the respirator on the face may create a possible obstruction in the airflow. This could possibly explain the high volumes reached, especially in Resusci Baby (Table 2). Based on the questionnaire results, the effect of feeling “short of breath” was no more prevalent than, e.g., when performing CPR in high altitudes where high-quality resuscitation can still be achieved [26]. Some of the students needed to adjust the position of the respirator on their faces during the 2 min cycle. However, this never resulted in unsuccessful rescue breath. There is a significant potential for further studies to gain the optimal efficacy and superior safety for rescuers in BLS mouth-to-mouth ventilation during the COVID-19 pandemic or anytime to reduce other aerosol transmittable diseases, as the transmission of viral particles is not prevented by the use of standardly available devices for mouth-to-mouth ventilation, such as a resuscitation shield or pocket mask. There are possible limitations in the reproducibility of our trial. The breathable self-sterilizing nanofiber respirators with accelerated copper that were used in the study might not be available worldwide, and their use in BLS training brings additional costs. However, with a rapidly evolving situation, the availability of these types of respirators can change quickly. Another limitation of the study could be the worldwide variable access to simulation mannequins. However, well-known, certified, and well-established mannequins for CPR simulation were chosen for this trial. The application of our results in clinical practice on real patients needs further prior studies. The results limitation could be intraindividual variability at assessing the effectivity on the Adult BLS mannequin because it depends strictly on the observer. Based on the observation of a 2 min cycle of BLS only, we could not see the eventual decrease of the quality of CPR over a longer time, as was observed in high altitudes [26]. Despite these possible limitations of the study, greater than 90% effectivity of mouth-to-mouth breathing during BLS was achieved together with superior airway protection (compared to the resuscitation mask). Currently, hand-only resuscitation can be performed (based on ERC guidelines) because the majority of lay rescuers are not willing to breath with the victim. The main reasons are the hygiene and infection issues. This could be solved with the FFP2 respirator. However, the application in clinical practice during BLS will require further investigation. Despite the discussed limitations of this study, effective ventilation of the mannequin with the FFP2 respirator was achieved. This approach possessed a comparable effectivity to mouth-to-mouth breathing together with superior safety. We have already successfully tested the use of respirators in first aid exercises in the autumn semester of 2021 in the undergraduate program at the Faculty of Medicine, Masaryk University, in Brno (763 students in the General Medicine and Dentistry program) without any complications. The possibility of reusing the respirator represents a great advantage. One piece was enough for the student for the whole semester of first aid training. We cannot predict how severe the pandemic will be in the future and what the COVID-19 measures will be like. Nevertheless, this technology will guarantee quality CPR training at our university in the future, including ventilation training, even during pandemic measures.

## 5. Conclusions

The breathable three-layer FFP2 nanofiber self-sterilizing respirator with accelerated copper for airway protection from SARS-CoV-2 can be effectively used for safe BLS simulation training, including mouth-to-mouth ventilation through the respirator with no need for extra manipulation, and could lead to a higher-quality BLS education during the pandemic.

## Figures and Tables

**Figure 1 children-09-01751-f001:**
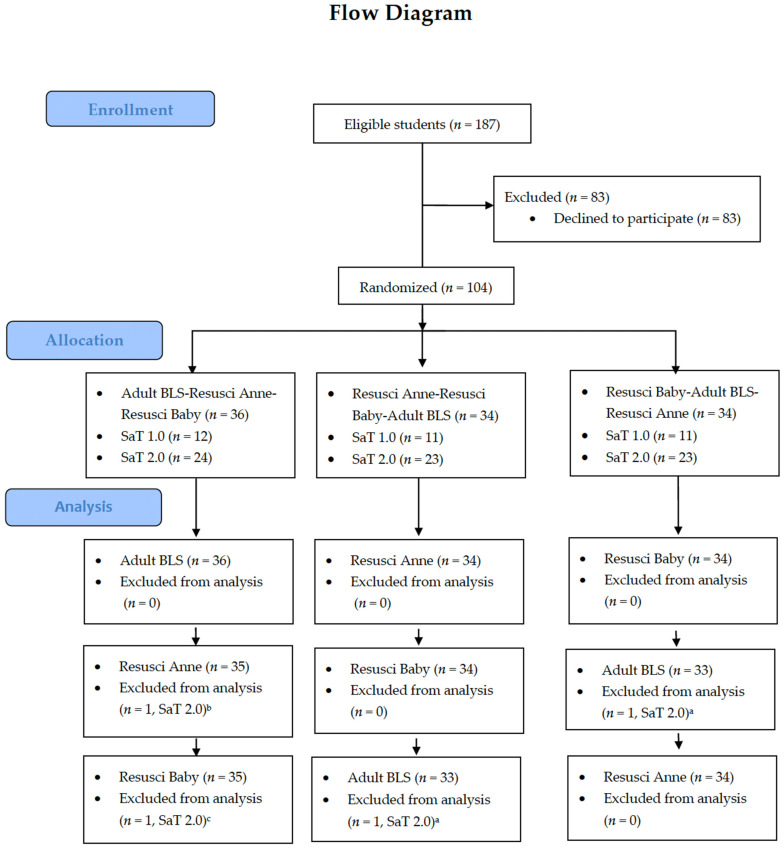
Flow chart showing the inclusion of study participants and allocation to three groups with sequential use of mannequins (*n* = 187). SaT: student as Teacher, BLS: Basic Life support; ^a^ Not correct technique of BLS (not pinched nose) in all attempts on the mannequin; ^b^ Not correct technique of BLS (not head tilt) in all attempts on the mannequin; ^c^ Not correct technique of BLS (no neutral head position) in all attempts on the mannequin.

**Figure 2 children-09-01751-f002:**
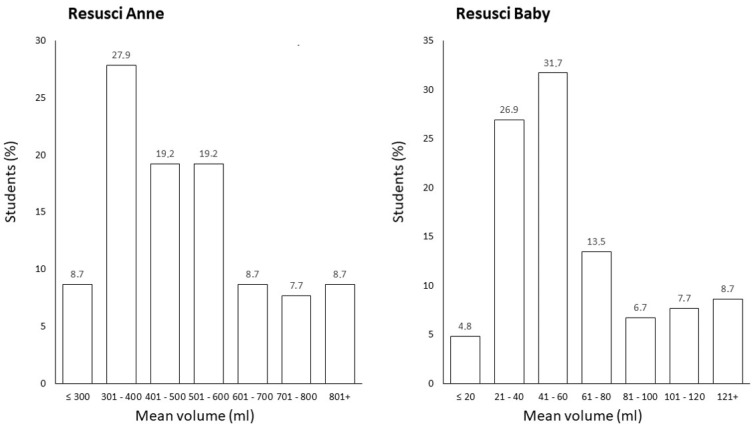
Mean volume of rescue breaths in all students (*n* = 104).

**Table 1 children-09-01751-t001:** Primary outcome—visible chest rise.

Valid Attempts for Rescue Breath	Visible Chest Rise *n* (%) ^1^	Mean of Attempts Per Subject ^2^ (*n* = 104)	Comparison	*p* ^3^
ADULT BLS (*n* = 981)	951 (96.9%)	96.9%	Resusci Anne vs. ADULT BLS	<0.001
Resusci Anne (*n =* 906)	822 (90.7%)	90.8%	Resusci Anne vs. Resusci Baby	<0.001
Resusci Baby (*n* = 1857)	1777 (95.7%)	95.8%	ADULT BLS vs. Resusci Baby	0.106

^1^ Computed from individual attempts, ^2^ Computed from aggregated results of individual subjects, ^3^ Statistical significances of differences based on individual attempts was computed using a generalized linear mixed effects model.

**Table 2 children-09-01751-t002:** Breath volume for Resusci Anne versus Resusci Baby.

	ADULT (Resusci Anne)		INFANT (Resusci Baby)	
	Attempts (%) ^1^ (*n* = 906)	Mean of Attempts Per Student ^2^ (*n* = 104)	Attempts (%) ^1^ (*n* = 1 857)	Mean of Attempts Per Student ^2^ (*n* = 104)
No volume	84 (9.3%)	9.2%	80 (4.3%)	4.2%
Low volume	262 (28.9%)	31.6%	295 (15.9%)	16.7%
Optimal volume	299 (33.0%)	32.5%	817 (44.0%)	42.1%
High volume	261 (28.8%)	26.7%	665 (35.8%)	36.9%

^1^ Computed from individual attempts, ^2^ Computed from aggregated results of individual subjects.

**Table 3 children-09-01751-t003:** Follow-up questionnaire (*n* = 104).

Question	1 (Easy)	2	3	4	5 (Impossible)
I felt mouth-to-mouth ventilation through a respirator was:	35 (33.7%)	51 (49%)	15 (14.4%)	3 (2.9%)	0
Mouth-to-mouth ventilation on the mannequin compared to ventilation without a respirator seemed to me: (3 = the same; if you cannot decide, do not mention any possibility) *:	1 (1.0%)	5 (5.0%)	46 (45.5%)	46 (45.5%)	3 (3.0%)

Results are shown as number *n* (%) of answers, * 3 students did not answer.

## Data Availability

Data supporting the reported results is available from the contact person: Martina Kosinova, Kosinova.Martina@fnbrno.cz; Tel.: +420-53223-4692.

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
