# Peer review of "Feasibility of Mouth-to-Mouth Ventilation through FFP2 Respirator in BLS Training during COVID-19 Pandemic (MOVERESP Study): Simulation-Based Study"

_children, 2022, doi:10.3390/children9111751_

Round 1

Reviewer 1 Report

Line 22 grammar: "the aim of the study was to assess the efficacy"

line 32: grammar should be "35.8%"

line 55-56: unclear purpose of sentence

line 70: should be our not out

line 146: I dont think invalid attempts should be excluded from analysis. For example say there were more invalid attempts due to the respiratory being awkward to get good seal. If you exclude those from analysis it could bias your results. Would recommend including all attempts in analysis. 

Line 156: Would add more info on power calculation. What did you set as estimate of effect, alpha, beta etc.

flow diagram: you have a c superscript that is not described below

table 1 primary outcome: I think a better control would be a comparison of visible chest rise from those who used respirator to those who did not. I think just comparing adult versus anne vs baby does not tell me much. For example say standard mouth to mouth had a effectivity of 98% compared to the 90% reported, that is a fairly big difference that may be worth knowing. I would recommend comparing to non respirator data either.

table 3: it seems a lot of people thought mouth to mouth ventilation with a respirator was much more difficult. 

Author Response

Dear reviewer,

Thank you for your comments on our article. We respond to all comments in the following text.

Line 22 grammar: "the aim of the study was to assess the efficacy"

Text corrected.

line 32: grammar should be "35.8%"

Text corrected.

line 55-56: unclear purpose of sentence

Text corrected.

line 70: should be our not out

Text corrected.

line 146: I don’t think invalid attempts should be excluded from analysis. For example, say there were more invalid attempts due to the respiratory being awkward to get good seal. If you exclude those from analysis it could bias your results. Would recommend including all attempts in analysis.

We understand your suggestion. We marked as invalid attempts those that did not meet the ERC guidelines 2015. However, the main reason is that if the inhalation was ineffective due to poor technique (e.g., not plugging the nose), the result could indicate that the respirator is responsible for the failure.

Line 156: Would add more info on power calculation. What did you set as estimate of effect, alpha, beta etc.

Information has been added in the text.

flow diagram: you have a c superscript that is not described below

Diagram corrected.

table 1 primary outcome: I think a better control would be a comparison of visible chest rise from those who used respirator to those who did not. I think just comparing adult versus anne vs baby does not tell me much. For example say standard mouth to mouth had a effectivity of 98% compared to the 90% reported, that is a fairly big difference that may be worth knowing. I would recommend comparing to non-respirator data either.

We agree with your argument. However, at the time of the study (Spring 2021), the epidemiological situation in the Czech Republic was very serious, the risk of transmitting the virus from the mannequin was extremely high, and our university had strict anti-vaccine measures in place. Therefore, it was not possible to obtain data without a respirator.

table 3: it seems a lot of people thought mouth to mouth ventilation with a respirator was much more difficult.

The mark 3 in the Follow-up questionnaire means that ventilation with the respirator had the same difficulty for the students compared to ventilation without the respirator. The mark 4 means it was a little bit difficult. We don’t interpret it as much more difficult.

Kind regards

Vaclav Vafek

Reviewer 2 Report

Dear Authors,

I would like to thank you for giving me the opportunity to review your paper. It is very well-structured research, and the emerging results will provide an advancement of the current knowledge about the optimal efficacy for the patient and superior safety for rescuers in BLS mouth-to-mouth ventilation during any pandemic on aerosol transmittable disease.

I have a strong agreement with the limitations of the study that reported inside of the paper. In my point of it will need many same studies so to apply breathable three layers FFP2 nanofiber self-sterilizing respirator with accelerated copper the in clinical practice on real patient. Most rescuers are not willing to breath with the victim mouth-to-mouth, because of infectious issues.

Sincerely

Author Response

Dear Reviewer, 

On behalf of our entire team, thank you for your review. We will try to take our project further. 

Sincerely 

Vaclav Vafek

Round 2

Reviewer 1 Report

My two biggest areas for improvement of study were exclusion of patient with poor technique and problem with adequate control (not using the ventilator). Neither problem was corrected. I understand not being feasible to use better control but at the very least I still feel that invalid attempts should not be excluded. Your argument that poor technique could affect results is valid, however I think the technique could be poor due to use of ventilator and excluding them introduces bias. At the least could it be run as a supplemental analysis to ensure the results stay the same.

Author Response

Dear reviewer,

Thank you for your review. My previous answer may have not been understandable. 
Our primary outcome was defined as the effectivity of mouth-to-mouth ventilations through the breathable respirator.  
If we had not excluded invalid attempts (poor ventilation technique not influenced by ventilator use), there would have been a significant violation of the trial methodology. We would not be following the primary outcome at this point (effectivity of the respirator), and the bias in results could be very substantial.

Kind regards 

Vaclav Vafek
